# Prevalence of soil-transmitted helminths and associated risk factors among primary school children in Kandahar, Afghanistan: A cross-sectional analytical study

Bilal Ahmad Rahimi[1,2]*, Najeebullah Rafiqi[3], Zarghoon Tareen[1], Khalil Ahmad Kakar[4], Mohammad Hashim Wafa[5], Muhammad Haroon Stanikzai[6], Mohammad Asim Beg[7], Abdul Khaliq Dost[3], Walter R. Taylor[8]

1 Department of Pediatrics, Faculty of Medicine, Kandahar University, Kandahar, Afghanistan, 2 Department of Pediatrics, Faculty of Medicine, Malalay Institute of Higher Education, Kandahar, Afghanistan,
3 Department of Surgery, Faculty of Medicine, Kandahar University, Kandahar, Afghanistan, 4 Department of Public Health, Faculty of Medicine, Malalay Institute of Higher Education, Kandahar, Afghanistan,
5 Department of Psychiatry, Faculty of Medicine, Kandahar University, Kandahar, Afghanistan,
6 Department of Public Health, Faculty of Medicine, Kandahar University, Kandahar, Afghanistan,
7 Department of Pathology and Laboratory Medicine, The Aga Khan University Hospital, Karachi, Pakistan,
8 Mahidol Oxford Tropical Medicine Clinical Research Unit (MORU), Mahidol University, Bangkok, Thailand

* drbilal77@yahoo.com

## Abstract

### Background

Soil-transmitted helminth (STH) infections are global health problem, especially in low-income countries. Main objectives of this study were to estimate the prevalence and intensity of STH and its risk factors among school children in Kandahar city of Afghanistan.

### Methodology/principal findings

This was a school-based cross-sectional analytical study, with data collected during eight-month-period (May–December, 2022) from 6- and 12-years old school children in Kandahar city, Afghanistan. All the stool samples were examined by saline wet mount method and Kato–Katz technique. Data were analyzed by using descriptive statistics, Chi square test, and multivariate logistic regression.

A total of 1275 children from eight schools of Kandahar city were included in this study. Mean age of these children was 8.3 years with 53.3% boys. The overall prevalence of any intestinal parasitic infection was 68.4%. The overall prevalence of STH infection was 39.1%, with *Ascaris lumbricoides* (29.4%) as the most prevalent STH species. Mean intensity of overall STH infection was 97.8. Multivariate logistic regression revealed playing barefoot (AOR 1.6, 95% CI 1.1–2.2), not washing hands after defecating and before eating (AOR 1.3, 95% CI 1.0–1.7), having untrimmed nails (AOR 1.4, 95% CI 1.1–1.8), and belonging to poor families (AOR 1.3, 95% CI 1.0–1.7) as the risk factors associated with the predisposition of school children for getting STH in Kandahar city of Afghanistan.

**Data Availability Statement:** All relevant data are within the paper and its Supporting Information files

**Funding:** WRT is partly is part funded by Wellcome under grant 220211. The funders had no role in study design, data collection and analysis, decision to publish, or preparation of the manuscript.

## Conclusions/significance

There is high prevalence of STH among school children of Kandahar city in Afghanistan. Most of the risk factors are related to poverty, decreased sanitation, and improper hygiene. Improvement of socioeconomic status, sanitation, and health education to promote public awareness about health and hygiene together with periodic mass deworming programs are better strategies for the control of STH infections in Afghanistan.

## Author summary

Soil-transmitted helminths (STH) are a group of intestinal parasites which globally infect more than 1.5 billion people, primarily in low- and middle-income countries. Approximately 260 million preschool-age and 654 million school-age children live in areas where the transmission of STH is very high. The risk of parasitic diseases is estimated to be very high in Afghanistan, but studies are rarely done on STH in Afghanistan. So, this study was conducted with main goals to estimate the prevalence and intensity of STH and its associated factors among primary school children in Kandahar city of Afghanistan. This study found that prevalence of STH is high among school children. Most of the factors associated with increased STH infections in Kandahar city school children are related to poverty, decreased sanitation, and improper hygiene. Improvement of socioeconomic status, sanitation, and health education to promote public awareness about health and hygiene together with periodic mass deworming programs are better strategies for the control of STH infections in Afghanistan.

## Introduction

Soil-transmitted helminths (STH) belong to a group of neglected tropical diseases, which occur primarily in low- and middle-income countries across tropical and subtropical regions, and disproportionately affect low-income communities [1]. Globally, World Health Organization (WHO) estimates more than 1.5 billion (24% of the world population) people are infected with STHs in Africa, Asia, and Latin America [1–3]. Over 260 million preschool-age and 654 million school-age children live in areas where STH transmission is very high [1].

Globally, the most prevalent STH is *Ascaris lumbricoides* (*A. lumbricoides*) (infecting approximately 1.2 billion people), followed by *Trichuris trichiura* (*T. trichiura*) (infecting nearly 795 million people) and hookworm (*Ancylostoma duodenale* and *Necator americanus*) which infects approximately 740 million people [4,5]. School age children are at high risk of being infected with STH. This could be due to the reason that these children are more exposed to contaminated soil when they play, walk barefoot, eat soil, and do not practice good personal hygiene [6]. Individuals with moderate-to-heavy intensity STH infections experience adverse health outcomes including diarrhea, abdominal pain, anemia, and impaired cognitive and physical development in children [5,7–11]. Deworming campaigns in different countries of the world have shown to improve nutritional status, cognition, and school performance in school-age children [12–14].

Afghanistan, for the last four decades, has been suffering from military and civil conflict. This, combined with natural disasters, has extremely weakened economic development [15]. The prevalence of STH is mostly unknown in Afghanistan. Diagnosis is mostly made on

clinical basis without any laboratory confirmation. The advent of Taliban government, decrease in international humanitarian aid, insecurity, and shortage of medical staff at all levels of the healthcare system hinders the implementation of epidemiological surveillance [16]. Healthcare system in Afghanistan is mostly dependent of international humanitarian aid [17]. The risk of parasitic diseases is estimated to be very high in Afghanistan [18].

Unfortunately, to our knowledge, only two studies have been conducted in Afghanistan to find the prevalence of STH among primary school children [19,20]. In 2003, World Health Organization (WHO) conducted a school-based survey with the help of Afghan ministry of public health and Afghan ministry of education. They collected fecal samples from 1,001 school children in four provinces of Afghanistan (Kabul, Kandahar, Nangarhar, and Farah) for soil-transmitted helminths (STH) [19]. In 2017, a follow-up survey was conducted among school children aged 8–10 years to provide an update on STH epidemiology in Kabul, Balkh, Herat, Nangarhar, and Kandahar provinces of Afghanistan [20]. In 2020, a community-based study among 1426 children was conducted to estimate the prevalence and associated factors of STH among children in Daman district of Kandahar province in Afghanistan. In this study, the overall prevalence of STH infection was 22.7%. Main risk factors associated with the predisposition of rural children for getting STH were not washing hands after defecating/before eating, living in mud house, walking barefoot, living in overcrowded house, and practicing open defecation [21]. In another hospital-based study in Afghanistan, 548 fecal samples were collected from the patients (both children and adults) with internal complaints who were admitted in two hospitals of Ghazni and Parwan provinces. More than one-third of these patients had intestinal helminths [16]. Main objectives of this school-based study were to estimate the prevalence and intensity of STH and its risk factors among school children in Kandahar city of Afghanistan.

## Methods

### Ethics statement

Written informed consents were taken from parents or guardians of all the participants prior to the study. Also, assent forms were given to the children for the participation. Information of the participants will not be disclosed. Ethical approval was taken from Kandahar University Ethics Committee (code number KDRU-EC-2022.19). For data collection, only children's initials were used. Prior to entering into the computer for analysis, the collected data was coded and de-identified. Also, to minimize the errors, data was double entered.

### Study design and study area

This was a school-based cross-sectional analytical study, conducted during eight-month-period (May–December, 2022) in Kandahar. Kandahar is a city, located in the south-west of Afghanistan. This city is located at an elevation of 1,010 meters. Kandahar is Afghanistan's second largest city after Kabul, with a population of about 614,118 people. All the schools of Kandahar city were selected for randomization using lottery method. After randomization, eight schools (four boys' schools and four girls' schools) were selected for the study. The four boys' school were Ahmad Shah Baba school, Mirwais Neka school, Zahir Shahi school, and Temor Shahi school. The four girls' schools were Zarghona Ana school, Malalai school, Nazo Ana school, and Aino Ana school.

## Study population and sample size calculation

Our source population was comprised of primary school children (both boys and girls) of class one to class five, with ages between 6 and 12 years. All those children were excluded from the study who received any anti-helminthic treatment in the previous three months before the commencement of the study, having chronic diseases, not able to provide stool samples, or their parents/guardians refused to participate in the study. If more than one children are coming from the same household, the statistics can be over represented in the household with many primary school age children. To avoid this issue, only one child per house was enrolled in this study.

The sample size and power calculations were performed in Epi Info version 7.2 (CDC, Atlanta, Georgia, USA). A 20% non-response rate was added. Our sample size was 1385 children. Among these children, parents/guardians of 32 (2.3%) children refused to take part in the study, 70 (5.1%) failed to submit their fecal samples, six (0.4%) had history of receiving anti-helminthic treatment in the last three months, while two (0.1%) had history of chronic disease (thalassemia). So, data was collected from 1275 children.

The total population of Kandahar city is 614,118 with primary school age population of 121,000 children. During 2022, there were a total of 17,950 (14.8% of primary school age population) primary school children with 11844 (66%) boys and 6,106 (34.0%) girls [22]. So, our sample size constitutes 7.1% of school population while 1.1% of all primary school-age children of Kandahar city.

## Sample collection and laboratory procedures

From all schools, children were selected using lottery method of randomization. A questionnaire was utilized in two local languages (Pashto and Dari) with questions regarding general characteristics, economic status, general sanitation and environmental conditions, and laboratory examination. Data were collected/recorded on paper forms by experienced and trained investigators. Prior to data collection, short briefings were given, during which the objectives and methods of the study were clearly informed to the children. The case record form and other materials were pretested before the actual data collection. Persons responsible for data collection were well trained on how to conduct the interview with children and their parents or care-takers and how to collect the stool samples. For stool sample collection, the children who agreed to participate in the study were provided with clean pre-labeled capped plastic container for stool collection along with instruction on correct placement of the stool into the containers. All the children were instructed to collect 100 mg of the stool samples. In the laboratory, saline wet mount method and Kato–Katz technique were used by expert laboratory technicians. Saline wet mount method was used for protozoa identification while Kato-Katz technique was used to detect and find the intensity of intestinal helminths. Kato-Katz thick smears were examined within one hour of its preparation to avoid over clearing of Hookworm eggs. The total number of eggs were expressed as eggs per gram (EPG) of stool. EPG was calculated to classify the infection intensity as light, moderate, and heavy infection. The severity of STHs infection is defined as light, moderate, and heavy intensity of infections, respectively, as follows *A. lumbricoides*: 1 to 4999 EPG, 5000 to 49999 EPG, and $\geq$50000 EPG; *T. trichiura*: 1 to 999 EPG, 1000 to 9999 EPG, and $\geq$10000 EPG; hookworm: 1 to 1999 EPG, 2000 to 3999 EPG, and $\geq$4000 EPG [23]. For quality control, 10% of the stool samples were randomly selected and examined by another experienced laboratory technician who was blinded for the previous test result.

## Data analysis

The data were entered into Microsoft Excel, cleaned, and imported to Statistical Package for the Social Sciences (SPSS) version 22 (Chicago, IL, USA) for statistical analysis. Descriptive analysis including frequency, mean, standard deviation (SD), and range was used to summarize demographic characteristics. Frequency and percentage were used to summarize categorical variables. Chi-square test (using crude odds ratio [COR]) was performed to assess the binary association between various categorical variables. All variables that were statistically significant in univariate analyses were assessed for independence in a multivariate logistic regression (using adjusted odds ratio [AOR]) to determine the factors associated with the predisposition of rural children for getting STH. A *P*-value of <0.05 was statistically significant.

## Results

### Socio-demographic, economic characteristics, and hygiene conditions

Among 1385 children, data was collected from 1275 children. So, our response rate was 92.1%. Among 679 (53.3%) boys, 190 (14.9%) were from Temor Shahi school, 176 (13.8%) from Mirwais Neka school, 161 (12.6%) from Ahmad Shah Baba school, and 152 (11.9%) from Zahir Shahi school. From 596 (46.7%) girls, 152 (11.9%) were from Malalai school, 150 (11.8) from Nazo Ana school, 148 (11.6%) from Aino Ana school, and 146 (11.5%) from Zarghona Ana school. Mean (SD) age of these children was 8.3 (1.8) years with 903/1275 (70.8%) belonging to poor families (daily income <2 USD per day) (Table 1).

### Prevalence of soil-transmitted helminths

The overall prevalence of STH infection was 39.1% (499/1275 children). *A. lumbricoides* (29.4%, 375/1275) was the most prevalent STH species, followed by *T. trichiura* (12.1%, 154/1275) and hookworm (8.7%, 111/1275). Prevalence of intestinal protozoa infection was 31.7% (404/1275) while prevalence of overall any intestinal parasitic infection was 68.4% (872/1275). *G. intestinalis* (*Giardia intestinalis*) was the most prevalent intestinal protozoa with a prevalence of 22.4%. Among the STH infected patients, single infection, double infection, and triple infections were present in 343/1275 (26.9%), 151/1275 (11.8%), and 5/1275 (0.4%) of the children, respectively. Among other intestinal parasites, *Hymenolepis nana* was the most prevalent (221/1275 [17.3%]) (Table 2 and Fig 1).

### The intensity of soil-transmitted helminths

The intensity of STHs infection was categorized based on the WHO classification thresholds using Kato-Katz thick smear method of parasites egg quantification expressed in eggs per gram (EPG) of stool. Mean intensity of overall STH infection was 97.8. The mean intensity of *A. lumbricoides*, *T. trichiura*, and hookworm spp. infection in our study was 105.4, 92.8, and 95.5, respectively. Light intensity of *A. lumbricoides*, *T. trichiura*, and hookworm spp. was present in 90.4% (339/375), 97.4% (150/154), and 95.5% (106/111) of the school children, respectively. Heavy intensity of STH was not observed in any of the children (Table 3).

### Risk factors of soil-transmitted helminths

In Chi-square test, statistically significant variables responsible for increased STH infection were children playing barefoot (COR 1.7, 95% CI [confidence interval] 1.2–2.3, and *p*-value 0.001), not washing hands after defecation and before eating (COR 1.5, 95% CI 1.1–1.9, and *p*-value 0.002), having untrimmed finger nails (COR 1.5, 95% CI 1.2–1.9, and *p*-value 0.001),

**Table 1. Socio-demographic and other characteristics of the study participasnts.**

| Variable | Total, n (%) (n = 1275) | Boys, n (%) (n = 679) | Girls, n (%) (n = 596) | *P*-value |
|---|---|---|---|---|
| STH present in stool | | | | |
| Yes | 499 (39.1) | 270 (39.8) | 229 (38.4) | 0.624 |
| No | 776 (60.9) | 409 (60.2) | 367 (61.6) | |
| Family daily income per person (in Afghanis) | | | | |
| ≥180 (≥2 USD) | 372 (29.2) | 210 (30.9) | 162 (27.2) | 0.142 |
| <180 (<2 USD) | 903 (70.8) | 469 (69.1) | 434 (72.8) | |
| Family size | | | | |
| <5 people | 297 (23.3) | 151 (22.2) | 146 (24.5) | 0.341 |
| ≥5 people | 978 (76.7) | 528 (77.8) | 450 (75.5) | |
| Mother's education | | | | |
| Literate | 149 (11.7) | 82 (12.1) | 67 (11.2) | 0.643 |
| Illiterate | 1126 (88.3) | 597 (87.9) | 529 (88.8) | |
| Father's education | | | | |
| Literate | 300 (23.5) | 137 (20.2) | 163 (27.3) | 0.003 |
| Illiterate | 975 (76.5) | 542 (79.8) | 433 (72.7) | |
| House construction | | | | |
| Concrete | 313 (24.5) | 198 (29.2) | 115 (19.3) | <0.001 |
| Mud | 962 (75.5) | 481 (70.8) | 481 (80.7) | |
| Source of drinking water | | | | |
| Safe | 1010 (79.2) | 509 (75.0) | 501 (84.1) | <0.001 |
| Unsafe | 265 (20.8) | 170 (25.0) | 95 (15.9) | |
| Toilet in school | | | | |
| Absent or not functional | 565 (44.3) | 179 (26.4) | 386 (64.8) | <0.001 |
| Present and functional | 710 (55.7) | 500 (73.6) | 210 (35.2) | |
| Washing hands after defecating/before eating | | | | |
| Yes | 419 (32.9) | 233 (34.3) | 186 (31.2) | 0.239 |
| No | 856 (67.1) | 446 (65.7) | 410 (68.8) | |
| Walking barefoot | | | | |
| Yes | 180 (14.1) | 99 (14.6) | 81 (13.6) | 0.613 |
| No | 1095 (85.9) | 580 (85.4) | 515 (86.4) | |
| Finger nail status | | | | |
| Untrimmed | 866 (67.9) | 467 (68.8) | 399 (66.9) | 0.485 |
| Trimmed | 409 (32.1) | 212 (31.2) | 197 (33.1) | |
| Habit of nail biting | | | | |
| Yes | 115 (9.0) | 60 (8.8) | 55 (9.2) | 0.808 |
| No | 1160 (91.0) | 619 (91.2) | 541 (90.8) | |
| Consumption of raw vegetables | | | | |
| Yes | 776 (60.9) | 266 (39.2) | 510 (85.6) | <0.001 |
| No | 499 (39.1) | 413 (60.8) | 86 (14.4) | |
| Habit of eating soil | | | | |
| Yes | 70 (5.5) | 50 (7.4) | 20 (3.4) | 0.002 |
| No | 1205 (94.5) | 629 (92.6) | 576 (96.6) | |
| Domestic animals present at home | | | | |
| Yes | 315 (24.7) | 219 (32.3) | 96 (16.1) | <0.001 |
| No | 960 (75.3) | 460 (67.7) | 500 (83.9) | |

N, Number; STH, Soil-transmitted helminth; USD, United States Dollar.

**Table 2. Species of intestinal parasitic infection among primary school children in Kandahar city.**

| Intestinal parasitic infection | Number (n = 1275) | Prevalence (%) |
|---|---|---|
| Overall any intestinal parasitic infection | 872 | 68.4 |
| Monoparasitism | 474 | 37.2 |
| Polyparasitism | 398 | 31.2 |
| Overall any STH infection | 499 | 39.1 |
| Single STH infection | 343 | 26.9 |
| Double STH infection | 151 | 11.8 |
| Triple STH infection | 5 | 0.4 |
| Overall any intestinal protozoa infection | 404 | 31.7 |
| STH | | |
| *Ascaris lumbricoides* | 375 | 29.4 |
| *Trichuris trichiura* | 154 | 12.1 |
| Hookworm spp. | 111 | 8.7 |
| Protozoa | | |
| *Giardia intestinalis* | 286 | 22.4 |
| Entamoeba spp. | 189 | 14.8 |
| Other intestinal parasites | | |
| *Hymenolepis nana* | 221 | 17.3 |
| Taenia spp. | 124 | 9.7 |
| *Enterobius vermicularis* | 50 | 3.9 |

n, Number; STH, Soil-transmitted helminth.

habit of nail biting (COR 1.5, 95% CI 1.0–2.2, and *p*-value 0.045), and belonging to poor families (COR 1.4, 95% CI 1.1–1.8, and *p*-value 0.009) (Table 4).

Multivariate logistic regression of the above-mentioned statistically significant variables revealed that playing barefoot (AOR 1.6, 95% CI 1.1–2.2, and *p*-value 0.006), not washing hands after defecating and before eating (AOR 1.3, 95% CI 1.0–1.7, and *p*-value 0.043), having untrimmed nails (AOR 1.4, 95% CI 1.1–1.8, and *p*-value 0.010), and belonging to poor families (AOR 1.3, 95% CI 1.0–1.7, and *p*-value 0.038) as the risk factors associated with the predisposition of school children for getting STH (Table 4). Hookworm infection and other STHs (ascariasis and trichuriasis) are different in the aspect of life cycle of the parasites. Therefore, we did the analysis for risk factors between these two different infections, but there were no statistically significant difference between them.

## Discussion

In this cross-sectional study, we studied 1275 school children during eight-month period (May–December, 2022). The overall prevalence of STH among school children of Kandahar city was 39.1% with mean intensity of 97.8. Main risk factors associated with STH among school children were playing barefoot, not washing hands after defecating and before eating, having untrimmed nails, and belonging to poor families.

In our study, prevalence of STH among school children was 39.1%. This prevalence is more than the studies conducted in Cameroon (1%) [24], Nepal (3.1%) [25], Indonesia (10.1%) [26], China (14.1%) [27], Tajikistan (32%) [28], and Malaysia (37%) [29]. Contrary, prevalence in our study is less than reported in studies from Philippines (84.7%) [30], Ethiopia (84.4%) [31], Nigeria (83.3%) [32], and India (75.6%) [33]. The prevalence variances observed in different parts of the world (and even different areas of the same country) are multifactorial; including

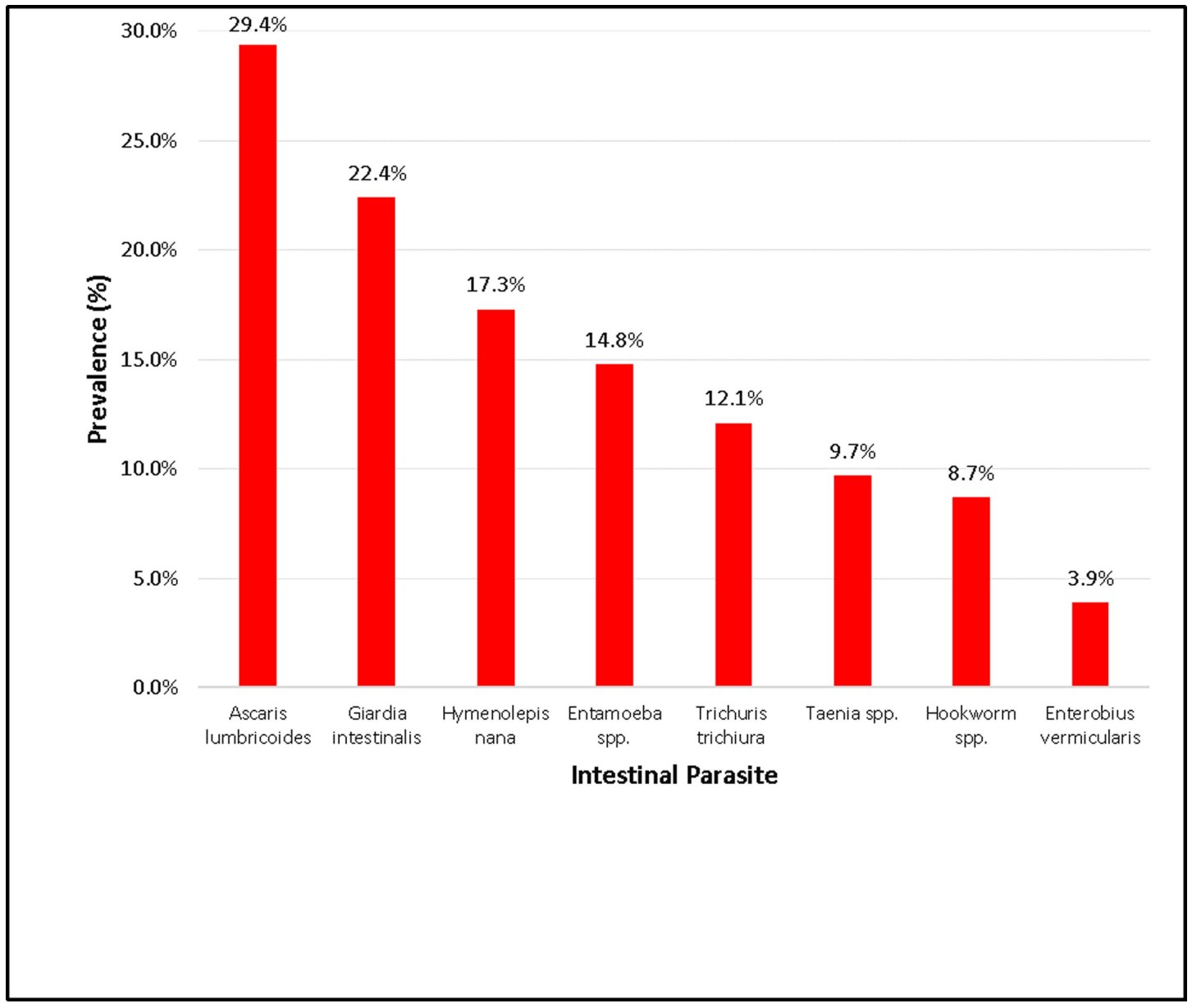

**Fig 1. Prevalence of intestinal parasites among school children in Kandahar city.**

differences in stool examination techniques, geographical location, time of study, type of study, age of study participants, culture, socio-economic status, literacy levels/occupations of the parents or guardians, food consumption habits, personal hygiene behaviors, and playing habits and facilities of children in and outside the school [34]. Table 5 compares the prevalence of STH infections in school children among our study and two other studies conducted in Kandahar city in 2003 and 2017. In our study, the prevalence of STH is lower but prevalence of hookworm is higher than other two studies conducted in Kandahar. In 2017 study, no moderate or heavy intensity STH infections were observed. In 2003 study, moderate-severe intensity was observed only in *A. lumbricoides* infection. Decreased STH prevalence (39.1%) in our study can be contributed to the collection of only one stool sample instead of the standard

**Table 3. Infection intensity of STHs among primary school children in Kandahar city.**

|  | Frequency, n (%) | EPG, mean (range) |
|---|---|---|
| Overall STH infection | n = 499 | 97.8 (2–13150) |
| *Ascaris lumbricoides* |  |  |
| Overall | n = 375 | 105.4 (5–13150) |
| Light intensity | 339 (90.4) | 103.6 (5–4370) |
| Moderate intensity | 36 (9.6) | 6380 (5600–13150) |
| Heavy intensity | 0 | 0 |
| *Trichuris trichiura* |  |  |
| Overall | n = 154 | 92.8 (3–1650) |
| Light intensity | 150 (97.4) | 90.3 (3–780) |
| Moderate intensity | 4 (2.6) | 1375 (1200–1650) |
| Heavy intensity | 0 | 0 |
| Hookworm spp. |  |  |
| Overall | n = 111 | 95.5 (2–2800) |
| Light intensity | 106 (95.5) | 92.4 (2–1680) |
| Moderate intensity | 5 (4.5) | 2250 (2180–2800) |
| Heavy intensity | 0 | 0 |

EPG, Eggs per gram; spp, Species.

three samples and Afghanistan ministry of public health (with the help of UN donor agencies) implementation of mass deworming interventions once in a year among school children throughout Afghanistan. Increased prevalence of hookworm in our study could be due to increased contact of school children to soil. In recent years, due to drought and decreased availability of ground water in Kandahar city, grass playgrounds have changed into soil playgrounds in nearly all schools.

The most prevalent STH in our study was *A. lumbricoides* (29.4%). The *A. lumbricoides* as the most common STH has also been reported in studies from Nepal (26.6%) [35], India (69.6%) [33], and Nigeria (75.6%) [36]. The reason behind the *A. lumbricoides* predominance could be associated to the long life of the female worm and has a fecundity rate of approximately 134,000–360,000 eggs per day for nearly 300 days. As a result, vast numbers of eggs are discharged into the human environment daily. Furthermore, the hard nature of these eggs to resist adverse environmental conditions more than other STHs can contribute to sustaining the transmission cycle for a longer period [37]. A study was conducted on 207 adults and 179 children visiting health facilities of Ghazni and Parwan provinces of Afghanistan. This study concluded that the most prevalent STH among children was *A. lumbricoides* (25.1% in Ghazni province and 10.8% in Parwan province) [16]. In 2003, a study conducted among school children of Kabul, Nangarhar, Farah, and Kandahar provinces of Afghanistan revealed that the most prevalent STH was *A. lumbricoides* with the prevalence of 41% (408/1001) in the school children of these four provinces of Afghanistan [19]. In 2017, another study conducted in Kabul, Balkh, Herat, Nangarhar and Kandahar provinces of Afghanistan showed that the most prevalent STH infection was the one with *A. lumbricoides* (25.7%) [20]. In 2020, a community-based study in rural children of Daman district in Kandahar Afghanistan also reported *A. lumbricoides* as the most prevalent STH species (18.7%) [21].

Aside from STH, the most prevalent (22.4%) intestinal parasite in our study was *G. intestinalis*. Similar results have been reported from another study in Afghanistan (14.5%) [16], Pakistan (28.9%) [38], Tajikistan (26.4%) [28], Kenya (26.1%) [39], and Peru (27.5) [40]. Evidence

**Table 4. Univariate analyses and logistic regression of risk factors associated with increased STH in primary school children.**

| Variable | Total, n (%) (n = 1275) | STH infection present | | COR (95% CI) | P-value | AOR (95% CI) | P-value |
|---|---|---|---|---|---|---|---|
| | | Yes, n (%) (n = 499) | No, n (%) (n = 776) | | | | |
| Gender | | | | | | | |
| Male | 679 (53.3) | 270 (39.8) | 409 (60.2) | 0.9 (0.8–1.2) | 0.624 | | |
| Female | 596 (46.7) | 229 (38.4) | 367 (61.6) | 1 | | | |
| Mother's education | | | | | | | |
| Literate | 149 (11.7) | 62 (41.6) | 87 (58.4) | 1 | 0.510 | | |
| Illiterate | 1126 (88.3) | 437 (38.8) | 689 (61.2) | 1.1 (0.8–1.6) | | | |
| Father's education | | | | | | | |
| Literate | 300 (23.5) | 122 (40.7) | 178 (59.3) | 1 | 0.535 | | |
| Illiterate | 975 (76.5) | 377 (38.7) | 598 (61.3) | 1.1 (0.8–1.4) | | | |
| Playing barefoot | | | | | | 1.6 (1.1–2.2) | |
| Yes (risky) | 180 (14.1) | 90 (50.0) | 90 (50.0) | 1.7 (1.2–2.3) | 0.001 | | 0.006 |
| No | 1095 (85.9) | 409 (37.4) | 686 (62.6) | 1 | | | |
| House construction | | | | | | | |
| Concrete | 313 (24.5) | 113 (36.1) | 200 (63.9) | 1 | 0.205 | | |
| Mud | 962 (75.5) | 386 (40.1) | 576 (59.9) | 0.8 (0.6–1.1) | | | |
| Source of drinking water | | | | | | | |
| Safe | 1010 (79.2) | 391 (38.7) | 619 (61.3) | 1 | 0.544 | | |
| Unsafe | 265 (20.8) | 108 (40.8) | 157 (59.2) | 0.9 (0.7–1.2) | | | |
| Family daily income per person (in Afghanis) | | | | | | | |
| ≥180 (≥2 USD) | 372 (29.2) | 125 (33.6) | 247 (66.4) | 1 | 0.009 | 1 | 0.038 |
| <180 (<2 USD) | 903 (70.8) | 374 (41.4) | 529 (58.6) | 1.4 (1.1–1.8) | | 1.3 (1.0–1.7) | |
| Family size | | | | | | | |
| <5 people | 297 (23.3) | 119 (40.1) | 178 (59.9) | 1 | 0.708 | | |
| ≥5 people | 978 (76.7) | 380 (38.9) | 598 (61.1) | 1.1 (0.8–1.4) | | | |
| Washing hands after defecating/before eating | | | | | | | |
| Yes | 419 (32.9) | 139 (33.2) | 280 (66.8) | 1 | 0.002 | 1 | 0.043 |
| No (risky) | 856 (67.1) | 360 (42.1) | 496 (57.9) | 1.5 (1.1–1.9) | | 1.3 (1.0–1.7) | |
| Finger nails status | | | | | | | |
| Untrimmed (risky) | 866 (67.9) | 365 (42.1) | 501 (57.9) | 1.5 (1.2–1.9) | 0.001 | 1.4 (1.1–1.8) | 0.010 |
| Trimmed | 409 (32.1) | 134 (32.8) | 275 (67.2) | 1 | | 1 | |
| Habit of nail biting | | | | | | | |
| Yes | 115 (9.0) | 55 (47.8) | 60 (52.2) | 1.5 (1.0–2.2) | 0.045 | 1.0 (0.6–1.6) | 0.944 |
| No | 1160 (91.0) | 444 (38.3) | 716 (61.7) | 1 | | | |
| Consumption of raw vegetables | | | | | | | |
| Yes | 776 (60.9) | 292 (37.6) | 484 (62.4) | 1 | 0.169 | | |
| No | 499 (39.1) | 207 (41.5) | 292 (58.5) | 0.9 (0.7–1.1) | | | |
| Habit of eating soil | | | | | | | |
| Yes | 70 (5.5) | 29 (41.4) | 41 (58.6) | 1 | 0.686 | | |
| No | 1205 (94.5) | 470 (39.0) | 735 (61.0) | 1.1 (0.7–1.8) | | | |
| Domestic animals present at home | | | | | | | |
| Yes | 315 (24.7) | 130 (41.3) | 185 (58.7) | 0.9 (0.7–1.2) | 0.371 | | |
| No | 960 (75.3) | 369 (38.4) | 591 (61.6) | 1 | | | |
| Toilet in school | | | | | | | |

(*Continued*)

**Table 4.** (Continued)

| Variable | Total, n (%) (n = 1275) | STH infection present | | COR (95% CI) | *P*-value | AOR (95% CI) | *P*-value |
|---|---|---|---|---|---|---|---|
| | | Yes, n (%) (n = 499) | No, n (%) (n = 776) | | | | |
| Absent or not functional | 565 (44.3) | 228 (40.4) | 337 (59.6) | 1.1 (0.9–1.1) | 0.427 | | |
| Present and functional | 710 (55.7) | 271 (38.2) | 439 (61.8) | | | | |

AOR, Adjusted Odds Ratio; CI, Confidence Interval; COR, Crude Odds Ratio; n, number; STH, Soil-transmitted helminth; USD, United States Dollar.

reveals that not only acute but chronic Giardiasis has an effect on the nutrition and health status of children [41]. Chronic intestinal protozoal infections are increasingly recognized as cause of undernutrition among children and have been proposed to be considered as neglected tropical diseases, causing morbidity in children comparable to infections caused by STHs [42].

Fortunately, severe intensity of STH infection was not observed in any child in this study. However, moderate intensity of *A. lumbricoides*, hookworm spp., and *T. trichiura* was present in 9.6%, 4.5%, and 2.6% of our study children, respectively. In study conducted during 2017 in Kandahar, no moderate-heavy intensity STH infections were observed [20]. This increase observed in our study could be contributed to the decrease in socio-economic status of people since the advent of Taliban in August 2021. Our result is higher than the WHO elimination target of STHs, which is defined as a <2% of moderate and heavy intensity due to STH infections [43]. Compared to our study, increased moderate-heavy intensity STH infections have been reported from Myanmar [44], Malaysia [45], Cameroon [46], and Rwanda [47]. Contrary, decreased moderate-heavy intensity STH infections have been reported from India [48], Ethiopia [37], and Nigeria [49].

Our study showed that children who were not washing hand after defecation and before eating were having statistically significant STH. Protective effects of handwashing have also been reported from Iran [50], Uzbekistan [51], China [27], India [52,53], Nepal [25], Lao [54], Indonesia [26], and Ethiopia [55]. In contrast, Wondemann and colleagues in Cuba reported a negative association between hand washing and infections [56]. This could be due to the lack

**Table 5. Prevalence of STH infections among school children in Kandahar city, in 2003, 2017, and 2022.**

| | 2003 [19] | 2017 [20] | 2022 (This study) |
|---|---|---|---|
| Sample size (n) | 257 | 452 | 1275 |
| Any STH (%) | 42.8 | 46.8 | 39.1 |
| *Ascaris lumbricoides* (%) | 37.4 | 45.5 | 29.4 |
| Light intensity | 91.7 | 100 | 90.4 |
| Moderate intensity | 8.3* | 0 | 9.6 |
| Heavy intensity | | 0 | 0 |
| *Trichuris trichiura* (%) | 7.8 | 1.4 | 12.1 |
| Light intensity | 100 | 100 | 97.4 |
| Moderate intensity | 0 | 0 | 2.6 |
| Heavy intensity | 0 | 0 | 0 |
| Hookworm spp. (%) | 0 | 0.5 | 8.7 |
| Light intensity | 0 | 100 | 95.5 |
| Moderate intensity | 0 | 0 | 4.5 |
| Heavy intensity | 0 | 0 | 0 |

* Moderate-heavy infection

STH, Soil-transmitted helminth.

of knowledge about transmission of the parasites or lack of awareness regarding health and hygiene habits among mothers [57].

In this study, living in poor families was a risk factor associated with increased STH infection among school children. Similar results have been reported in other studies [38,58,59]. It could be due to the reason that poor families have very little access to clean water, sanitation, and hygiene.

Our study revealed that walking barefoot is a risk factor for having increased STH infection. Similar results have been reported from Nepal [60], Thailand [61], Indonesia [26], Malawi [62], Ethiopia [63], and Kenya [64]. On the other hand, studies conducted in India did not notice any association between footwear usage and STH [65,66]. Walking barefoot is especially a risk factor for hookworms, as their larvae in the soil can penetrate into unbroken skin. Although walking barefoot is not directly related to infections of other helminths, but it indirectly leads to the infection when child touches the contaminated feet and eat with unwashed hands afterwards [67].

Having untrimmed nails was also a risk factor of increased STH infection in our study. This result is in accordance with researches reported from Ethiopia [68–70] and Thailand [71]. This could be due to the outdoor playing habits of school children on poor sanitation areas, which results in contamination of their hands. Dirt under fingernails may harbor different stages of parasites, which can be ingested during food eating and nail-biting or thumb sucking [69,71].

## Limitations

There were some limitations in our study. We obtained only one fecal sample instead of the ideal three consecutive samples due to unavailability of fund, the level of cooperation, and response of the caretakers and children. This might underestimate the real burden of STH. Also, this study did not focus on molecular assays and other techniques that are best to estimate the prevalence of STHs and differentiate different species of hookworm. We did not get data of clinical symptoms and underlying diseases of the children, which can be confounding factors for STH. Additional studies should be performed in different parts of the country.

## Conclusion

Based on the results of our study, more than one-third of the primary school children were infected with at least one STH species. This indicates that STHs are still a health problem among primary school children in Kandahar. Main risk factors associated with the predisposition of school children for getting STH were playing barefoot, children from poor families, children not washing hand after defecation/before eating, and children with untrimmed finger nails.

We recommend that improvement of socioeconomic status, sanitation, and health education to promote public awareness about health and hygiene. Periodic mass deworming programs are crucial for the control of STH infections in Afghanistan. Besides albendazole, praziquantel or niclosamide should be added to the deworming program, to cover *H. nana* and Taenia spp. too. Meanwhile, government and international donor agencies in Afghanistan should help in improving socio-economic status of the people through provision of basic facilities such as piped water, electricity, good housing, and proper toilets.

## Supporting information

**S1 File. Microsoft excel file with some of the data of this study.**
(XLSX)

## Acknowledgments

We present our highest and sincere thanks to the authorities of Faculty of Medicine, Kandahar University, Kandahar Province Directorate of Education, and parents of the children of the school children in Kandahar City. We are also very thankful of the data collectors and all our study participants. For the purpose of Open Access, the author has applied a CC BY public copyright licence to any Author Accepted Manuscript version arising from this submission.

**Availability of data and materials**

The authors confirm that all data underlying the findings are fully available without restriction. All relevant data are within the paper and its Supporting Information files.

## Author Contributions

**Conceptualization:** Bilal Ahmad Rahimi, Najeebullah Rafiqi, Zarghoon Tareen, Khalil Ahmad Kakar, Mohammad Hashim Wafa, Muhammad Haroon Stanikzai, Mohammad Asim Beg, Abdul Khaliq Dost, Walter R. Taylor.

**Data curation:** Bilal Ahmad Rahimi, Najeebullah Rafiqi, Zarghoon Tareen, Khalil Ahmad Kakar.

**Formal analysis:** Bilal Ahmad Rahimi, Mohammad Hashim Wafa, Muhammad Haroon Stanikzai, Mohammad Asim Beg.

**Investigation:** Mohammad Hashim Wafa, Muhammad Haroon Stanikzai, Abdul Khaliq Dost.

**Methodology:** Bilal Ahmad Rahimi, Muhammad Haroon Stanikzai, Mohammad Asim Beg, Abdul Khaliq Dost, Walter R. Taylor.

**Project administration:** Bilal Ahmad Rahimi, Najeebullah Rafiqi, Mohammad Hashim Wafa, Muhammad Haroon Stanikzai.

**Resources:** Bilal Ahmad Rahimi, Najeebullah Rafiqi, Zarghoon Tareen, Khalil Ahmad Kakar, Mohammad Hashim Wafa, Muhammad Haroon Stanikzai, Mohammad Asim Beg, Abdul Khaliq Dost, Walter R. Taylor.

**Software:** Bilal Ahmad Rahimi, Najeebullah Rafiqi, Zarghoon Tareen, Khalil Ahmad Kakar, Mohammad Hashim Wafa.

**Supervision:** Bilal Ahmad Rahimi, Najeebullah Rafiqi.

**Validation:** Khalil Ahmad Kakar, Mohammad Hashim Wafa, Muhammad Haroon Stanikzai.

**Visualization:** Zarghoon Tareen, Khalil Ahmad Kakar, Abdul Khaliq Dost.

**Writing – original draft:** Bilal Ahmad Rahimi, Najeebullah Rafiqi, Walter R. Taylor.

**Writing – review & editing:** Bilal Ahmad Rahimi, Najeebullah Rafiqi, Zarghoon Tareen, Khalil Ahmad Kakar, Mohammad Hashim Wafa, Muhammad Haroon Stanikzai, Mohammad Asim Beg, Abdul Khaliq Dost, Walter R. Taylor.

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
