## [Decision Letter · Decision Letter 0]

2 Aug 2023

Dear Prof. Dr Rahimi,

Thank you very much for submitting your manuscript "Prevalence of soil-transmitted helminths and associated risk factors among primary school children in Kandahar, Afghanistan: A cross- sectional analytical study." for consideration at PLOS Neglected Tropical Diseases. As with all papers reviewed by the journal, your manuscript was reviewed by members of the editorial board and by several independent reviewers. The reviewers appreciated the attention to an important topic. Based on the reviews, we are likely to accept this manuscript for publication, providing that you modify the manuscript according to the review recommendations. 

Sincerely,

Peter Steinmann, Ph.D.

Academic Editor

Francesca Tamarozzi

Section Editor

Reviewer's Responses to Questions

**Key Review Criteria Required for Acceptance?**

**Methods**

-Are the objectives of the study clearly articulated with a clear testable hypothesis stated?

-Is the study design appropriate to address the stated objectives?

-Is the population clearly described and appropriate for the hypothesis being tested?

-Is the sample size sufficient to ensure adequate power to address the hypothesis being tested?

-Were correct statistical analysis used to support conclusions?

-Are there concerns about ethical or regulatory requirements being met?

Reviewer #1: LAB TECHNIQUES

It is not clear what lab technique was applied to detect which parasite. 

Was the saline wet mount method used for protozoa and the Kato-Katz for helminths?

SCHOOL ENROLMENT

Please clarify the primary school enrolment rate (%) in Kandahar, and comment on whether children included in the study can be considered as representative the school-age population of the city

Reviewer #2: Yes, but some information is missing. Please refer to general comments section.

**Results**

-Does the analysis presented match the analysis plan?

-Are the results clearly and completely presented?

-Are the figures (Tables, Images) of sufficient quality for clarity?

Reviewer #1: Please clarify what % of the entire schoolchildren population of Kandahar (or what % of schools) were included in the survey (to understand how representative the sample was).

Reviewer #2: Yes, but some needs to be revised. Please refer to general comments section.

**Conclusions**

-Are the conclusions supported by the data presented?

-Are the limitations of analysis clearly described?

-Do the authors discuss how these data can be helpful to advance our understanding of the topic under study?

-Is public health relevance addressed?

Reviewer #1: These comments cover both sections "Discussion" and "Conclusions"

(1) Please add comments on intensity of STH infections and its implications on morbidity. 

(2) Please add information on intensity of STH infections to Table 5 and draw a comparison: it looks like that in 2017 no moderate-intensity infections were found but in your survey you do find some moderate-intensity infections.

(3) Please add details on the current status of STH deworming activities carried out in Afghanistan in general and in Kandahar more specifically.

(4) Although the title of the paper implies that its focus is on STH, data on intestinal protozoa and on other helminths were nevertheless collected and presented. It would be useful to include these in the discussion and draw a comparison with information taken from previously-implemented surveys 

(5) The "Discussion" section contains some repetition of information already presented in the "Introduction" - please streamline as necessary

Reviewer #2: Yes

**Editorial and Data Presentation Modifications?**

Reviewer #1: Please refer to above comments

Reviewer #2: (No Response)

**Summary and General Comments**

Reviewer #1: A descriptive study contributing to the body of evidence on STH in Afghanistan. Some limitations: 

(1) The survey is focused on a single province, Kandahar. 

(2) It looks like there is some overlapping with a paper published by the same first author in 2022 (with very similar title), although this was focused on a single district; in light of this, sentences such as "The risk factors of STH among school children are unknown in Afghanistan" (Background) do not seem to be appropriate. 

(3) Discussion on STH should focus more on intensity of infection (see comments above)

Reviewer #2: The authors demonstrated the prevalence of STH and associated risk factors among PSAC in Afghanistan. Although there is limitation in analyzing one sample per individual, it can provide evidence for high prevalence and associated risk factors that need interventions. This article has strength in including large number of PSACs and investing risk factors in the same individuals, but more detailed descriptions and analysis are needed.

These are some inquiry and suggestions. 

1. Authors stated that children who received any anti-helminthic treatment in the previous three months were excluded, but the number excluded by this criteria was not described. On page 9, only those whose parents refused or who failed to submit their fecal samples were described. And it is hard to know previous treatment history or presence of chronic disease before the investigation. Please describe how it was investigated and how many were excluded by this criteria. If the previous treatment was based on the stool exam or symptom based, this could be another reason why this study shows lower prevalence resulting from selection bias.

2. (Table 1) The percentage next to the should be described the other way, describing proportions within the boys or girls. The current way of description is affected by the total number of boys and girls included, and cannot be used to describe differences in proportions between boys and girls.

3. Hookworm infection and other STHs (ascariasis and trichuriasis) are different in the aspect of life cycle of the parasites. If there was difference in the result of risk factor analysis between these two different infections, please describe it. Even if not, please state it in the manuscript. For instance, it would be interesting to see if barefoot is also associated with ascariasis or trichuriasis in this study.

4. It seems this study include a number of families with more than 5 people. Can authors identify household information? If many of the individuals are coming from the same household, the statistics can be over represented in the household with many PSACs. Please indicate if authors have household information, and if there are substantial number, it would be beneficial to provide analysis with household data.

Minor inquiry and suggestions.

1. Typical KK methods include a number of eggs found in a slide multiplied by 24. How many slides were made in one sample, and how authors could get 2,3, or 5 as EPGs? 

2. It seems that children were questioned, not guardians were according to the description in methods. Please indicate if the guardians were questioned.

PLOS authors have the option to publish the peer review history of their article (what does this mean?). If published, this will include your full peer review and any attached files.

Reviewer #1: No

Reviewer #2: Yes: Hyun Beom Song

Figure Files:

Data Requirements:

Please note that, as a condition of publication, PLOS' data policy requires that you make available all data used to draw the conclusions outlined in your manuscript. Data must be deposited in an appropriate repository, included within the body of the manuscript, or uploaded as supporting information. This includes all numerical values that were used to generate graphs, histograms etc.. For an example see here: http://www.plosbiology.org/article/info:doi%2F10.1371%2Fjournal.pbio.1001908#s5.

Reproducibility:

References

---

## [Editor Report · Decision Letter 1]

21 Aug 2023

Dear Prof. Dr Rahimi,

We are pleased to inform you that your manuscript 'Prevalence of soil-transmitted helminths and associated risk factors among primary school children in Kandahar, Afghanistan: A cross- sectional analytical study.' has been provisionally accepted for publication in PLOS Neglected Tropical Diseases.

Best regards,

Peter Steinmann, Ph.D.

Academic Editor

Francesca Tamarozzi

Section Editor

---

## [Editor Report · Acceptance letter]

7 Sep 2023

Dear Prof. Dr Rahimi,

We are delighted to inform you that your manuscript, "Prevalence of soil-transmitted helminths and associated risk factors among primary school children in Kandahar, Afghanistan: A cross- sectional analytical study.," has been formally accepted for publication in PLOS Neglected Tropical Diseases.

Best regards,

Shaden Kamhawi

co-Editor-in-Chief

Paul Brindley

co-Editor-in-Chief
